**Subject Category:**
Biology (whole organism)

evolution/ecology/behaviour

sensory drive, habitat choice, haplochromine, ecological speciation, colour vision

**Author for correspondence:**
Daniel Mameri
e-mail: dmameri@isa.ulisboa.pt

# Visual adaptation and microhabitat choice in Lake Victoria cichlid fish

Daniel Mameri[1,2], Corina van Kammen[2,3],
Ton G. G. Groothuis[2], Ole Seehausen[4,5]
and Martine E. Maan[2]

[1]CEF – Forest Research Centre, School of Agriculture, University of Lisbon, Lisbon, Portugal
[2]Groningen Institute for Evolutionary Life Sciences (GELIFES), University of Groningen, Groningen, The Netherlands
[3]Van Hall Larenstein University of Applied Sciences, Leeuwarden, The Netherlands
[4]Department of Fish Ecology and Evolution, Eawag Center for Ecology, Evolution and Biogeochemistry, Kastanienbaum, Switzerland
[5]Institute of Ecology and Evolution, University of Bern, Bern, Switzerland

DM, 0000-0002-8942-3792; TGGG, 0000-0003-0741-2174;
OS, 0000-0001-6598-1434; MEM, 0000-0003-1113-8067

When different genotypes choose different habitats to better match their phenotypes, genetic differentiation within a population may be promoted. Mating within those habitats may subsequently contribute to reproductive isolation. In cichlid fish, visual adaptation to alternative visual environments is hypothesized to contribute to speciation. Here, we investigated whether variation in visual sensitivity causes different visual habitat preferences, using two closely related cichlid species that occur at different but overlapping water depths in Lake Victoria and that differ in visual perception (*Pundamilia* spp.). In addition to species differences, we explored potential effects of visual plasticity, by rearing fish in two different light conditions: broad-spectrum (mimicking shallow water) and red-shifted (mimicking deeper waters). Contrary to expectations, fish did not prefer the light environment that mimicked their typical natural habitat. Instead, we found an overall preference for the broad-spectrum environment. We also found a transient influence of the rearing condition, indicating that the assessment of microhabitat preference requires repeated testing to control for familiarity effects. Together, our results show that cichlid fish exert visual habitat preference but do not support straightforward visual habitat matching.

# 1. Introduction

In heterogeneous environments, individuals may disperse to (micro)habitats that best match their phenotype and thereby

increase their ecological performance (i.e. 'matching habitat choice') [1]. This behaviour may dissipate natural selection for local adaptation but, if it causes habitat segregation, it may contribute to genetic differentiation between (micro)habitats and ultimately speciation [2,3]. Here, we investigate this process in the context of sensory drive, testing whether divergent visual phenotypes preferentially seek out alternative visual environments.

Visual systems adapt rapidly, responding to environmental challenges associated with foraging, predator avoidance and (sexual) communication [4–6]. This is particularly well documented in visually heterogeneous aquatic environments [7–9]. In Lake Victoria (East Africa), divergent visual adaptation is associated with speciation in the genus *Pundamilia* [6,10,11]. Sympatric *Pundamilia* species, with either blue or red male nuptial coloration, inhabit different (but overlapping) depth ranges and thereby experience different light environments: the blue species tend to inhabit shallow waters, receiving broad-spectrum light, while the red species tend to inhabit deeper waters with red-shifted light conditions [6]. The two species differ in opsin gene sequence (light-sensitive proteins in the eye; [6]) and opsin gene expression [12], and in visual response to blue and red light [9], corresponding to the difference in visual habitat. Recent work suggests that at least some of these differences are adaptive: when raising the fish in artificial light conditions that mimic shallow and deep habitats, both species survive best in their own natural light condition [13].

In this study, we test whether differences in visual sensitivity between blue and red *Pundamilia* species cause different visual habitat preferences. We expect that when given a choice, individuals will disperse from a suboptimal visual environment to one that better matches their visual system phenotype [1]. We address the contributions of genetic effects (i.e. species differences) and developmental effects (i.e. light regime during rearing).

If genetic differences determine visual habitat preference, we predict that individuals of either species prefer the light regime that is closest to the one their populations are adapted to. In addition to genetic differences, however, developmental plasticity may contribute to variation in visual sensitivity [14]. To explore this and to assess the causal relationship between visual sensitivity and habitat preference, we manipulate visual development by raising the fish under different light conditions. We have previously shown that these light treatments induce changes in opsin expression in *Pundamilia* [15]. We therefore predict that the light regime during development influences visual habitat preference as well.

In addition to individuals from the blue and the red species, we also include laboratory-bred first-generation interspecific hybrids. These are not expected to exert a genetically determined preference for either light environment, because their visual system presumably has intermediate characteristics, and they survive equally well in both environments [13]. Thus, we expect that if environment-induced plastic changes in visual sensitivity influence preference, the effect will be most pronounced in hybrids.

# 2. Material and methods

## 2.1. Pundamilia species

*Pundamilia pundamilia* (Seehausen *et al.*, 1998) and *Pundamilia nyererei* (Witte-Maas & Witte, 1985) co-occur at several rocky islands in southeastern Lake Victoria. Males are distinguished by their nuptial coloration; *P. pundamilia* are blue/grey, whereas *P. nyererei* are bright red and yellow. Females of both species are yellow/grey [16]. At each location, the sympatric species tend to have different depth distributions—the blue species occurs in shallow waters while the red species extends to greater depths [6,16]. Due to the high turbidity of Lake Victoria, this means that the red species inhabit a light environment largely lacking short-wavelength light [6,9,17]. In line with this, the red species carry LWS alleles (long-wavelength-sensitive opsin) that confer a more red-shifted sensitivity, compared to the allele that dominates in the blue species [6,12], and it has a greater behavioural sensitivity to long-wavelength light [9].

Here, we used laboratory-bred offspring from wild-caught fish, collected in 2010 and 2014 at Python Islands in southern Lake Victoria [6]. Until recently, all red populations were thought to belong to *P. nyererei* and all blue populations to *P. pundamilia*. However, the populations in the western and southern Mwanza Gulf (including Python Islands) represent a separate speciation event and are referred to as *P.* sp. 'pundamilia-like' and *P.* sp. 'nyererei-like' [10].

## 2.2. Test subjects

All tested fish were F1 and F2 sub-adults (aged 2–12 months; mean ± s.e. 4.5 ± 0.3; electronic supplementary material, table S1). We chose to use young fish to: (i) minimize the effects of differential mortality until testing

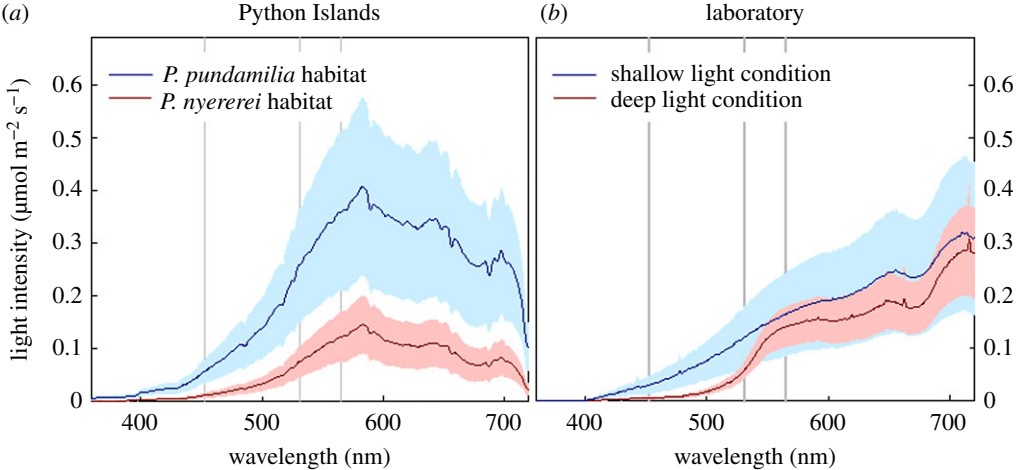

**Figure 1.** Light conditions at Python Islands and in the laboratory. (*a*) Downwelling irradiance in the natural habitats of *P.* sp. 'pundamilia-like' (0.5–2 m depth; blue curve) and *P.* sp. 'nyererei-like' (0.5–5 m depth, red curve). (*b*) Downwelling irradiance in the 'shallow' (blue curve) and 'deep' (red curve) light treatments in the laboratory. Curves represent averages of multiple measurement series with standard errors. Grey vertical lines indicate the maximum sensitivity of the three main photoreceptors of *Pundamilia*.

[13], and (ii) reduce the incidence of territoriality and aggression [16]. To control visual development of the test fish, they were removed as eggs from mouthbrooding females around the time of hatching (approx. 5 days after fertilization). Each family was then split, divided equally over two light treatments that mimicked the natural light environments experienced by *P.* sp. 'pundamilia-like' (shallow water, 0–2 m) and *P.* sp. 'nyererei-like' (deeper water, 0–5 m) at Python Islands [6] (figure 1; details below). They remained in these conditions throughout the experimental period and only experienced the other condition during the experiment itself. Thus, fish reared in broad-spectrum light were never exposed to the red-shifted light condition (and vice versa), until testing. Fish were maintained in full-sibling groups in 6.5 l tanks (27 × 17 × 14 cm) at 25 ± 0.5°C and 12 L : 12 D, and fed with fish flakes 6 days per week.

We used 120 fish: 40 *P.* sp. 'nyererei-like' (F1, mean age ≈ 10 ± 1 weeks), 40 *P.* sp. 'pundamilia-like' (F1, mean age ≈ 18 ± 8 weeks) and 40 hybrids (F1 and F2, mean age ≈ 27 ± 4 weeks, with a 50% P and 50% N genetic background). Fish were tested in groups of fixed composition (rather than individually, to reduce stress), with four siblings from the same light treatment. We used five sibling groups for each species and for the hybrids, from each condition, generating a total of 30 groups. For a list of experimental groups, see electronic supplementary material, table S1. Fish were not individually recognized and only group-level data were recorded. To allow group identification, fish were clipped in the dorsal or tail fin (most fish were housed with untested siblings). Behaviour was quantified with BORIS software [18]. All procedures were conducted between March and June 2016 and followed approved animal care protocols (RUG IACUC 6205B; AVD105002016464).

## 2.3. Light conditions

Light treatments were based on the natural light environments experienced by *P.* sp. 'pundamilia-like' and *P.* sp. 'nyererei-like' at Python Islands (figure 1; electronic supplementary material, figure S1). We measured downwelling irradiance (in μmol m$^{-2}$ s$^{-1}$) at Python Islands using a BLK-C-100 spectrophotometer and F-600-UV–VIS–SR optical fibre with CR2 cosine receptor (Stellar-Net, FL). Measurements were collected in 0.5 m depth increments down to 5 m depth. In each measurement series, we took a minimum of two irradiance spectra at each depth and used the average for further analysis. We collected four independent measurement series (20 and 26 May, 4 and 5 June 2010, between 9.00 and 11.00 h).

For each measurement series separately, we then estimated the light environments experienced by *P.* sp. 'pundamilia-like' and *P.* sp. 'nyererei-like', by calculating a weighted average of the spectra at each depth, using as a weighting factor the depth distribution of each species as reported in [6]. The average of the four resulting species-specific light spectra was mimicked in the laboratory by halogen lights (Philips Halogen Masterline ES, 30 and 35 W) filtered with a green filter (#243, LEE Filters, Andover, UK). In the shallow light condition, blue lights (Paulmann 88090 ESL Blue Spiral 15 W) were added to create a broad light spectrum. In the deep light condition, short-wavelength light was

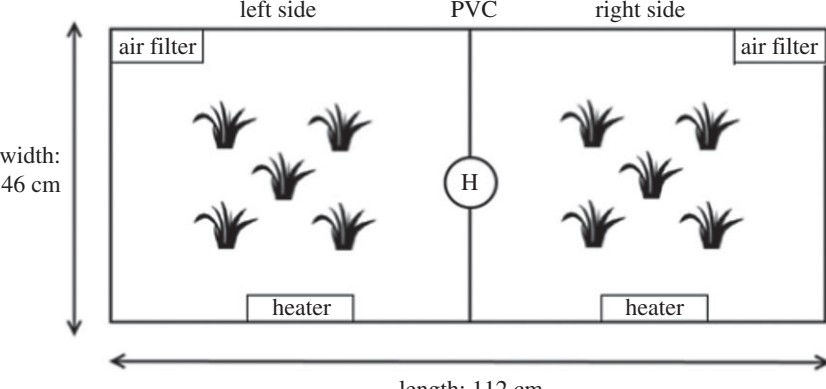

**Figure 2.** Experimental tank. A PVC sheet divided the tank into two equally sized compartments. A hole (H) in the divider allowed the fish to perceive the other light environment and cross from one side to the other. To make both sides equally attractive, each was enriched with sand, plastic plants, an air filter and a heater (25 ± 0.5°C).

reduced by adding a yellow filter (LEE, #15). The resulting downwelling irradiance was measured with the same equipment as in the field (averaged across two tanks for each light condition, and four positions in each tank).

To verify the resemblance between natural and laboratory light conditions for the wavelengths that are most relevant for the *Pundamilia* visual system, we estimated the proportion of incident light captured by the three main photopigments of *Pundamilia*, for both laboratory and field spectra. Electronic supplementary material, figure S2 shows that in both field and laboratory conditions, the 'deep' light condition generates lower SWS (short-wavelength-sensitive) and higher LWS light capture than the 'shallow' light condition, with laboratory conditions slightly exaggerating the differences.

We did not attempt to mimic also the light intensity differences between habitats. At Python Islands, light intensity in the deeper *P.* sp. 'nyererei-like' habitat is about 34% of that in the shallow *P.* sp. 'pundamilia-like' habitat (figure 1). We did not adjust the experimental light spectra to reproduce this difference (intensity in deep condition was 70% of that in the shallow condition).

## 2.4. Experimental set-up and procedures

The experimental tank (112 × 46 × 41 cm; 211 l) was divided into two equally sized compartments by an opaque PVC sheet, with a semicircular hole of 10 cm diameter at the bottom to allow movement between sides (figure 2; electronic supplementary material, figure S3). One side of the tank had the shallow light condition (broad-spectrum) and the other one the deep light condition (red-shifted spectrum), which could be reversed.

Prior to each trial, chemical cues (scent of Chironomidae larvae) were spread in both sides of the tank to stimulate exploration. Two group members were introduced on each side. Observation time started as soon as one individual crossed to the other side or looked through the hole to the other side. For 1 h, we recorded the number of fish on each side. As a measure of activity, we also counted the number of times individuals crossed between sides. Trials were considered successful if at least four crossings were recorded. Groups were excluded if unsuccessful twice. Fish were returned to housing tanks after testing. All groups were tested twice, with approximately two weeks between repeats. Light environments were switched between tank sides after the first repeat. After analysing the first two repeats, we submitted *P.* sp. 'pundamilia-like' and *P.* sp. 'nyererei-like' groups (but not hybrid groups) to a third repeat to increase statistical power for testing species differences.

## 2.5. Data analyses

We calculated the proportion of time spent on the side of the tank with shallow light conditions (summed for the individuals in a group) as a measure of preference, ranging from 0 to 1, and then applied an arcsine transformation to improve data distribution. This preference score was fit into a linear mixed model, with the genetic group (*P.* sp. 'pundamilia-like', *P.* sp. 'nyererei-like' and hybrids), rearing environment (shallow and deep), age and activity as fixed effects. Random effects included repeat number, nested in fish group, nested in family (some groups came from the same family—see

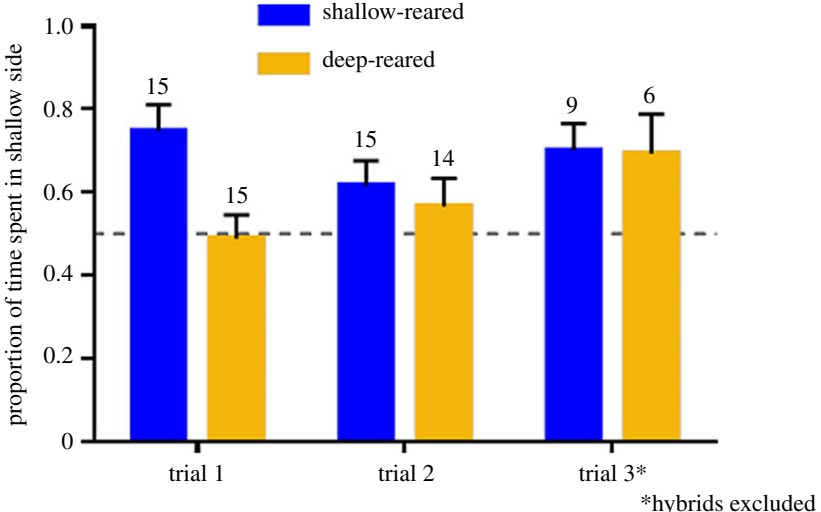

**Figure 3.** Fish spent more time in the shallow light condition, but deep-reared fish expressed no preference in the first repeat. Bars are means with standard errors; blue bars: shallow-reared fish; yellow bars: deep-reared fish. Numbers above bars indicate the number of test groups.

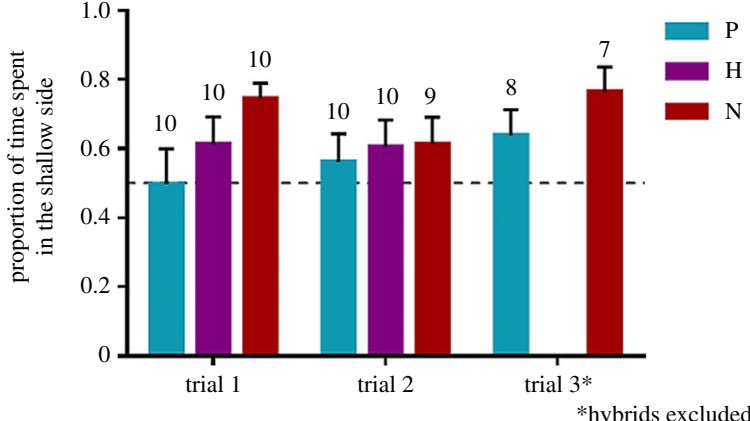

**Figure 4.** Visual habitat preference in *P.* sp. 'pundamilia-like' ('P', blue), hybrids ('H', purple) and *P.* sp. 'nyererei-like' ('N,' red), in repeats 1, 2 and 3. Bars are means with standard errors; numbers above bars indicate the number of test groups.

electronic supplementary material, table S1). Significance of each variable was tested using Satterthwaite's ANOVA, and the minimum adequate model was obtained by removing non-significant variables following a stepwise approach. All analyses were conducted in R [19].

# 3. Results

Of the 80 trials, 75 were successful. Overall, fish preferred the blue-shifted (shallow) light environment (figure 3; mean $\pm$ s.e.: $61 \pm 21\%$). Preference was significantly influenced by the environment fish were reared in ($F_{1;42.638} = 6.947$, $p = 0.012$; figure 3; electronic supplementary material, table S2). However, the difference between rearing groups was only seen in the first repeat: while shallow-reared fish expressed a consistent preference for the shallow environment, deep-reared fish initially had no preference but developed a preference for shallow in subsequent repeats (figure 3; electronic supplementary material, figure S4). Indeed, a separate model, with 'repeat' as a single fixed effect, was significant in deep-reared fish ($F_{1;23.618} = 7.382$, $p = 0.012$) but not in the shallow-reared fish ($F_{1;33.290} = 0.706$; $p = 0.407$; electronic supplementary material, table S3). We found no differences in preference between genetic groups ($F_{2;9.273} = 1.284$; $p = 0.322$; figure 4; electronic supplementary material, table S2). Repeating the analyses without hybrids also did not reveal differences between *P.* sp. 'pundamilia-like' and *P.* sp. 'nyererei-like' groups ($F_{1;8.097} = 0.265$, $p = 0.620$).

Fish activity (mean $\pm$ s.e. $= 34.6 \pm 3.37$ crossings per trial) was also significant in explaining light preference ($F_{1;69.742} = 8.383$, $p = 0.005$; electronic supplementary material, table S2): more active groups expressed weaker preferences for the shallow light condition. To explore this further, we calculated preference strength (deviation from 0.5, irrespective of the chosen light condition) and found that more active groups expressed weaker preferences overall ($F_{1;74} = 14.387$, $p < 0.001$; electronic supplementary material, figure S5 and table S4). Fish activity did not significantly differ between species ($F_{2;469} = 1.052$, $p = 0.391$), rearing conditions ($F_{1;66.372} = 0.442$, $p = 0.508$), repeats ($F_{1;57-132} = 0.679$, $p = 0.413$) or age classes ($F_{1;44.708} = 1.005$, $p = 0.322$).

## 4. Discussion

Matching habitat choice can evolve in response to selection for improving performance in heterogeneous environments [1]. Here, we investigated this phenomenon in two closely related cichlid species with divergent visual system characteristics, testing the hypothesis that individuals should preferentially reside in the light environment that mimics their natural habitat. Such preferences could contribute to genetic differentiation between populations, particularly during the early stages of adaptive divergence.

Contrary to expectation, the shallow water-dwelling *P.* sp. 'pundamilia-like' and the deeper-dwelling *P.* sp. 'nyererei-like' did not differ in visual habitat preference. Instead, we found an overall preference for the broad-spectrum light condition, mimicking shallow waters. This is surprising, given that opsin genotype is subject to divergent selection between these species, as evidenced by genetic signatures of divergent selection (on the long-wavelength-sensitive opsin gene, LWS [6,11]) and differences in survival between light environments in captivity [13]. In other fish, preferences for light conditions that maximize performance have been demonstrated [20,21]. Possibly, our fish did not have enough opportunity to evaluate their performance in the two environments: while we added food cues to stimulate exploration, we did not provide an actual reward that could generate vision-dependent variation in performance. Matching habitat choice may be most pronounced when it generates a substantial advantage [22]. Therefore, future experiments should provide an opportunity for such an advantage, in the form of e.g. food reward or social interaction.

For *P.* sp. 'pundamilia-like', the observed preference for the broad-spectrum light condition was expected, as it corresponds to its natural habitat. However, we expected *P.* sp. 'nyererei-like' to prefer the red-shifted light condition. *P.* sp. 'nyererei-like' also occurs in shallow water, but it is most abundant in deeper waters [6,11]. Possibly, this depth distribution does not reflect the preferred environment for this species but rather emerges from ecological interactions such as competition and predation. Predation risk by birds is higher in shallow waters, which might affect especially the brightly coloured males of *P.* sp. 'nyererei-like' [23]. Both of these factors, competition and predation, were absent in our experiment.

We found a transient effect of the rearing light environment: in the first repeat, deep-reared fish did not express a preference for the shallow light condition. Developmental effects on visual habitat preference have been observed in some fish species but not others (e.g. Australasian snapper prefer light intensities that match their rearing environment [20], but Coho salmon prefer darker backgrounds even when raised in bright illumination [24]). In this study, it seems that familiarity with the rearing environment may have suppressed exploration of the unfamiliar one [21]. To explore this further, we also assessed preference in 15 min blocks within the first repeat (electronic supplementary material, figure S6). We did not observe that deep-reared fish gradually spent more time in the unfamiliar environment (shallow) in the course of this first repeat, suggesting that habituation requires longer or more frequent exposure. Either way, this finding entails a caution for future studies: testing individuals only once may poorly estimate behavioural preferences.

We have previously shown that the light treatments we used here induce changes in opsin expression in *Pundamilia* [15]: deep-reared fish express less short-wavelength-sensitive opsin (SWS2a) and more long-wavelength-sensitive opsin (LWS). Yet, we did not observe a sustained effect of the rearing environment on preference. Possibly, the induced changes in opsin expression (5–15%) were too subtle to cause behavioural effects. More extreme rearing environments can generate larger changes (e.g. [25]) that would potentially influence visual habitat preference in a more persistent way.

An alternative explanation for the lack of a sustained effect of the rearing light treatment is that the differences in opsin expression were erased during the experimental trials, as fish were exposed to both light conditions during this time. It is unknown how quickly opsin expression can change in response to altered light conditions. Studies in killifish [26] and cichlids [27] suggest that changes can occur within a

few days, but most studies have used much longer exposure times (weeks to months). We expect that exposure of 1 h is too short to induce significant changes, but we cannot rule this out.

Our light treatments differed in both spectral composition and intensity. Therefore, we cannot establish which of these aspects, or a combination of both, was responsible for the observed variation in visual habitat preference. Previous studies [20,24] have recorded fish preferences for darker or brighter environments, but these employed larger differences between habitats (ranging from twofold to 20-fold) than we used in the present study (light intensity in the red-shifted condition was 70% of that in the broad-spectrum condition). Independently manipulating both spectral composition and light intensity is feasible and would constitute a logical next step.

To conclude, we find evidence that *Pundamilia* cichlid fish exert significant preference for visual habitat, preferring broad-spectrum over darker, red-shifted light conditions. Species differences in visual traits and habitat in the wild do not translate into differences in preference. Light conditions during development do influence preference, but only in the short term. We conclude that our results do not support a simple role of vision-mediated matching habitat choice in *Pundamilia* cichlids.

Permission to carry out fieldwork. No fieldwork was required for this study, as tested fish were bred in captivity (F1 and F2 offspring; see electronic supplementary methods for details).

Ethics. All procedures followed approved animal care protocols (RUG IACUC 6205B; AVD105002016464).

Data accessibility. Data are provided in the electronic supplementary material. Data available from the Dryad Digital Repository at: https://doi.org/10.5061/dryad.fd4d84h [28].

Authors' contributions. M.E.M., O.S. and T.G.G.G. designed the study; D.M. and M.E.M. developed the experimental set-up; D.M. and C.v.K. collected the data. D.M. analysed the data; D.M. and M.E.M. wrote the manuscript, with contributions from O.S. and T.G.G.G. All authors gave final approval for publication.

Competing interests. The authors declare that they have no competing interests.

Funding. This article was funded by Swiss National Science Foundation (SNSF PZ00P3-126340 to M.E.M.), The Netherlands Foundation for Scientific Research (NWOVENI 863-009-005 to M.E.M.) and the Erasmus+ programme (scholarship 15/SMT/2015 to D.M.).

Acknowledgements. We acknowledge the Tanzanian Commission for Science and Technology for research permission and the Tanzanian Fisheries Research Institute for hospitality and facilities. We thank M. Haluna, M. Kayeba, E. Ripmeester and O. Selz for help in the field, S. Veenstra, B. Verbeek, A. Taverna and E. Schaeffer for taking care of the fish in the laboratory, and S. Wright and G. Nunes for help with fish care and experimental design. We thank P. Edelaar and an anonymous reviewer for comments that improved the manuscript.

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
