## [Reviewer comments · Royal Society Open Science]

Review History

RSOS-181876.R0 (Original submission)

Review form: Reviewer 1

Is the manuscript scientifically sound in its present form?

No

Are the interpretations and conclusions justified by the results?

No

Is the language acceptable?

Yes

Is it clear how to access all supporting data?

Yes

Do you have any ethical concerns with this paper?

No

Have you any concerns about statistical analyses in this paper?

No

Recommendation?

Major revision is needed (please make suggestions in comments)

Comments to the Author(s)

The authors aimed at testing whether divergent visual sensitivity affects visual habitat preferences. They also aimed at testing whether manipulating visual sensitivity by rearing fish under two different light conditions affects the species preferences for lighting. To answer these questions, the authors used two closely related Lake Victoria cichlids that differ in their depth preferences – one inhabits deeper red-shifted water than the other does. Such depth preference has been previously suggested to correspond to differences in the visual sensitivity of the species. The authors conclude that in contrast to their hypothesis, fish did not show obvious preference to the light conditions that mimicked their typical natural habitat.

Major concerns

In the wild, the species that occupies deeper water is exposed to a red-shifted spectrum but also to an overall lower light intensity. Without knowing the intensity of the two experimental light regimes, it is hard to conclude what aspect of the two light conditions drove the preference to the 'broad spectrum' condition. For example, it might be possible that fish actually preferred the brighter or dimmer light condition.

What are the characteristics of the experimental light environment in terms of spectrum and intensity? Were light intensities matched to those encountered in the natural habitats? It would be useful to present these data relative to the lighting regimes encountered in the wild. What were the estimated quantum catches of the various opsin types expressed in each of the species?

One of the mechanisms of the presumed divergence in visual sensitivity between the species is differential opsin gene expression. However, the authors do not provide any evidence on whether opsin expression differed between the tested species. Is it possible that all fish exhibited similar gene expression as a result of their exposure to the same light condition prior to the beginning of the experiment? How fast such plasticity in opsin gene expression could be?

The authors state (lines 55-57) that they manipulated visual development by raising the fish under different light conditions. However, they provide no evidence for the capacity of such manipulation. In other words, did the different light conditions actually affected the visual sensitivity, and if so, how?

Minor concerns

Lines 39-43 – It is not clear what are the differences between *P. pundamilia* and *P. sp.* 'pundamilia-like'. The same is also true for *P. nyererei*. Please clarify.

Line 46 – 'They' presumably refers to the two species, but this isn't clear enough.

Line 51 – Please elaborate on how the causality between visual sensitivity and habitat preferences is being determined.

Line 74 – 'only group-level data was recorded' should be 'only group-level data were recorded'.

Line 74-75 – What was the light regime before fish were exposed to the experimental light regimes?

Line 86-88 – “After analysing the first two trials, we submitted *P. sp.* ‘pundamilia-like’ and *P. sp.* ‘nyererei-like’ to a third trial to increase statistical power for testing species differences.” Why not simply say that groups were tested three times. Did the third trial differ from the two previous ones? Actually, only when I reached Figure 2 I realized that the third trial differed from the rest in that hybrids were not tested. It might be better to clarify this issue up front.

Line 99 – There is inconsistency in the use of the term ‘trial’. Line 87 says that the species were subjected to a third trial. However, in line 99, it is indicated that ‘75 of the 80 trials were successful’. Please clarify.

Line 119 – ‘nor trials’ should probably be replaced with ‘or trials’.

Review form: Reviewer 2 (Pim Edelaar)

Is the manuscript scientifically sound in its present form?

Yes

Are the interpretations and conclusions justified by the results?

Yes

Is the language acceptable?

Yes

Is it clear how to access all supporting data?

Yes

Do you have any ethical concerns with this paper?

No

Have you any concerns about statistical analyses in this paper?

Yes

Recommendation?

Accept with minor revision (please list in comments)

Comments to the Author(s)

In this ms the authors investigate whether genetic background and rearing environment affect preference for light environment in two closely related cichlid species which might have diverged in the wild due to distinct habitat preferences as related to their visual systems. Hence, the study tests a clear speciation scenario, one which so far has almost not been tested. I therefore found the study interesting, relevant and timely.

The experimental design is adequate, as are the analysis. Sample size is perhaps somewhat small (as in many behavioural studies), which may mask small effects. It might be worth to discuss the issue of lack of power, and whether any observed trends at least go in the predicted direction or not. I also have a few suggestions for improvement, or clarification, of analyses.

The main predictions are mostly not confirmed. While I’m aware that conceptually negative

results are just as good as positive ones and would not want to promote a publication bias in favour of positive results, there is always the possibility that the hypotheses are in fact correct, but the experimental test was not. You discuss that perhaps reward for correct habitat choice was not sufficiently in place, which is important. You discard the difference in light intensity across habitats which was not implemented, for reasons that I didn't find so convincing. And then there may be another 1000 factors that are important for the fish, but that we are not aware of and therefore were not (correctly) manipulated. Perhaps in your paper, mention this possibility – that your experimental test might just not have created the ecological setting that does play out in the field, and therefore that the results are limited. This is fine, that is how science works, and perhaps the hypotheses are indeed wrong. But if you had achieved confirmatory results, the ecological relevance would probably not have to be questioned to the same extent.

L 32: add: and thereby increase their ecological performance

L 60: is there nothing known about (co)dominance in expression?

L 61: as written now, your predictions appear exclusive, which cannot be: they prefer their natural light regime (L 58), and they prefer their rearing light regime. Why not simply state that you will test if, and to what extent, genetic background and rearing regime affect habitat preference? (And then mention the directions re. these effects).

L 68: so did you allocate entire families to treatment, or did you split families between treatments (a stronger design)?

L 71: is the range 0-5 m the natural range? How do you expose fish to the light of a depth range? It is in the supplement (weighting by depth use), but briefly clarify this here.

L 75: this is important, since you might then be testing for novelty avoidance (neofobia), or novelty attraction (neofilia). OK, you discuss this later (L 144).

L 92: I was wondering why you didn't use a binomial model, but there must be temporal autocorrelation in your timing events, so it is hard to know how many independent time event you would have, so the transformation seems indeed better.

L 94: activity is an interesting one. Why would activity influence habitat use by itself? More likely, more active fish are exposed to the different environments more, and therefore can choose better (which is probably why you demanded a minimum of four transitions). Or reverse, more active fish move between environments more, therefore not choosing. Either way, I think it is the interactions between activity*species, and activity*rearing environment, that are most interesting, not the main effect of activity.

L 94: you don't include side ("is the shallow light condition on the left or right") as a fixed factor?

L 94: what is trial number, and why does this have replicate values (required for a random effect)? Each trial supposedly is unique? If you refer to trial order, then this is not correct – fit it as a fixed effect.

L 96: you mention AIC, but then give p-values, which doesn't make much sense. And what are minimal adequate models – the outcome of stepwise elimination? Why not just give the results for the full models, testing all effects via loglikelihood ratio tests?

L 105: this results suggests you fitted trial number as a fixed effect, not as a random effect? And you tested an interaction, but you never specified before you would test any interactions – in "data analyses" indicate which interactions you included, and why.

L 105: 5 decimals for a p-value is way beyond its reliability: I think 3 decimals is precision enough.

L 116: I think this is a convoluted way to test an interaction?

L 132: genetic signatures?

L 138: and perhaps this depends also on competition? It is interesting that in the wild, the deep fish also use shallow waters, but not the reverse, mimicking the overall preference you detected here for shallow conditions perhaps? Might deep fish only use the deep when outcompeted by shallow fish in shallow waters (any field data to test this density- or frequency-dependence)? If so, and as you already mentioned, perhaps your design was not entirely appropriate to test your hypothesis, as reward was not included.

L 149: well, not quite – behavioural preference may depend on several factors (e.g. familiarity versus performance), and may shift over time.

L 151: no hypothesis of why all fish preferred shallow light conditions?

L 154: maybe add that future experiments should try to increase the importance of relative ecological performance across test environments, or something like that (i.e. greater reward/punishment).

Fig. 2: there seems to be a pattern here, I'm surprised it doesn't come out? But it actually goes against your prediction, right? The deep species prefers shallow water more?

Supplementary materials:

L 21: there is a strong correlation between age and genetic background. Could the inclusion of age in your models have removed the effect of genetic background?

L 60: I'm not so convinced that average difference in light intensity isn't important because cloud cover variation creates greater changes. Sun set creates even greater changes, but daylight spectrum differences still matter according to you. Perhaps acknowledge in your paper that this aspect was not manipulated, and might change the results. Someone else may try and replicate your study (in other species perhaps), taking this into account. Just as the reward issue.

I hope these comments help to further improve an already fine paper,

Pim Edelaar.

Decision letter (RSOS-181876.R0)

17-Dec-2018

Dear Mr Mameri,

The editors assigned to your paper ("Visual adaptation and microhabitat choice in Lake Victoria cichlid fish") have now received comments from reviewers. We would like you to revise your paper in accordance with the referee and Associate Editor suggestions which can be found below (not including confidential reports to the Editor). Please note this decision does not guarantee eventual acceptance.

Please submit a copy of your revised paper before 09-Jan-2019. Please note that the revision deadline will expire at 00.00am on this date. If we do not hear from you within this time then it will be assumed that the paper has been withdrawn. In exceptional circumstances, extensions may be possible if agreed with the Editorial Office in advance. We do not allow multiple rounds of revision so we urge you to make every effort to fully address all of the comments at this stage. If deemed necessary by the Editors, your manuscript will be sent back to one or more of the original reviewers for assessment. If the original reviewers are not available, we may invite new reviewers.

- Data accessibility

If you wish to submit your supporting data or code to Dryad (<http://datadryad.org/>), or modify your current submission to dryad, please use the following link:
<http://datadryad.org/submit?journalID=RSOS&manu=RSOS-181876>

- Competing interests

- Authors' contributions

- Acknowledgements

- Funding statement

Please note that Royal Society Open Science charge article processing charges for all new submissions that are accepted for publication. Charges will also apply to papers transferred to Royal Society Open Science from other Royal Society Publishing journals, as well as papers submitted as part of our collaboration with the Royal Society of Chemistry (<http://rsos.royalsocietypublishing.org/chemistry>). If your manuscript is newly submitted and subsequently accepted for publication, you will be asked to pay the article processing charge, unless you request a waiver and this is approved by Royal Society Publishing. You can find out more about the charges at <http://rsos.royalsocietypublishing.org/page/charges>. Should you have any queries, please contact openscience@royalsociety.org.

on behalf of Dr Kristina Sefc (Associate Editor) and Kevin Padian (Subject Editor)
openscience@royalsociety.org

Associate Editor's comments (Dr Kristina Sefc):

Dear authors,
Your manuscript has been seen by two reviewers. The RSOS format allows you to expand the manuscript, which will help in addressing their concerns in your revision. Reviewer 1 is concerned that light intensity, not only spectrum, may have played a role in the experiment, and asks how experimental light conditions corresponded to natural conditions. Additionally, more background on cichlid opsin gene expression may help to clarify some issues. Following reviewer 2, the discussion could delve deeper into the various factors (beyond the tested ones) that might have influenced experimental habitat choice.
Best regards, Kristina Sefc

Comments to Author:

Reviewers' Comments to Author:
Reviewer: 1

Comments to the Author(s)

The authors aimed at testing whether divergent visual sensitivity affects visual habitat preferences. They also aimed at testing whether manipulating visual sensitivity by rearing fish under two different light conditions affects the species preferences for lighting. To answer these questions, the authors used two closely related Lake Victoria cichlids that differ in their depth preferences – one inhabits deeper red-shifted water than the other does. Such depth preference has been previously suggested to correspond to differences in the visual sensitivity of the species. The authors conclude that in contrast to their hypothesis, fish did not show obvious preference to the light conditions that mimicked their typical natural habitat.

Major concerns

In the wild, the species that occupies deeper water is exposed to a red-shifted spectrum but also to an overall lower light intensity. Without knowing the intensity of the two experimental light regimes, it is hard to conclude what aspect of the two light conditions drove the preference to the 'broad spectrum' condition. For example, it might be possible that fish actually preferred the brighter or dimmer light condition.

What are the characteristics of the experimental light environment in terms of spectrum and intensity? Were light intensities matched to those encountered in the natural habitats? It would be useful to present these data relative to the lighting regimes encountered in the wild. What were the estimated quantum catches of the various opsin types expressed in each of the species?

One of the mechanisms of the presumed divergence in visual sensitivity between the species is differential opsin gene expression. However, the authors do not provide any evidence on whether opsin expression differed between the tested species. Is it possible that all fish exhibited similar gene expression as a result of their exposure to the same light condition prior to the beginning of the experiment? How fast such plasticity in opsin gene expression could be?

The authors state (lines 55-57) that they manipulated visual development by raising the fish under different light conditions. However, they provide no evidence for the capacity of such manipulation. In other words, did the different light conditions actually affected the visual sensitivity, and if so, how?

Minor concerns

Lines 39-43 – It is not clear what are the differences between *P. pundamilia* and *P. sp.* 'pundamilia-like'. The same is also true for *P. nyererei*. Please clarify.

Line 46 – 'They' presumably refers to the two species, but this isn't clear enough.

Line 51 – Please elaborate on how the causality between visual sensitivity and habitat preferences is being determined.

Line 74 – 'only group-level data was recorded' should be 'only group-level data were recorded'.

Line 74-75 – What was the light regime before fish were exposed to the experimental light regimes?

Line 86-88 – "After analysing the first two trials, we submitted *P. sp.* 'pundamilia-like' and *P. sp.* 'nyererei-like' to a third trial to increase statistical power for testing species differences." Why not simply say that groups were tested three times. Did the third trial differ from the two previous ones? Actually, only when I reached Figure 2 I realized that the third trial differed from the rest in that hybrids were not tested. It might be better to clarify this issue up front.

Line 99 – There is inconsistency in the use of the term 'trial'. Line 87 says that the species were subjected to a third trial. However, in line 99, it is indicated that '75 of the 80 trials were successful'. Please clarify.

Line 119 – 'nor trials' should probably be replaced with 'or trials'.

Reviewer: 2

Comments to the Author(s)

In this ms the authors investigate whether genetic background and rearing environment affect preference for light environment in two closely related cichlid species which might have diverged in the wild due to distinct habitat preferences as related to their visual systems. Hence, the study tests a clear speciation scenario, one which so far has almost not been tested. I therefore found the study interesting, relevant and timely.

The experimental design is adequate, as are the analysis. Sample size is perhaps somewhat small (as in many behavioural studies), which may mask small effects. It might be worth to discuss the issue of lack of power, and whether any observed trends at least go in the predicted direction or not. I also have a few suggestions for improvement, or clarification, of analyses.

The main predictions are mostly not confirmed. While I'm aware that conceptually negative results are just as good as positive ones and would not want to promote a publication bias in favour of positive results, there is always the possibility that the hypotheses are in fact correct, but the experimental test was not. You discuss that perhaps reward for correct habitat choice was not sufficiently in place, which is important. You discard the difference in light intensity across habitats which was not implemented, for reasons that I didn't find so convincing. And then there may be another 1000 factors that are important for the fish, but that we are not aware of and therefore were not (correctly) manipulated. Perhaps in your paper, mention this possibility – that your experimental test might just not have created the ecological setting that does play out in the field, and therefore that the results are limited. This is fine, that is how science works, and perhaps the hypotheses are indeed wrong. But if you had achieved confirmatory results, the ecological relevance would probably not have to be questioned to the same extent.

L 32: add: and thereby increase their ecological performance

L 60: is there nothing known about (co)dominance in expression?

L 61: as written now, your predictions appear exclusive, which cannot be: they prefer their natural light regime (L 58), and they prefer their rearing light regime. Why not simply state that you will test if, and to what extent, genetic background and rearing regime affect habitat preference? (And then mention the directions re. these effects).

L 68: so did you allocate entire families to treatment, or did you split families between treatments (a stronger design)?

L 71: is the range 0-5 m the natural range? How do you expose fish to the light of a depth range? It is in the supplement (weighting by depth use), but briefly clarify this here.

L 75: this is important, since you might then be testing for novelty avoidance (neofobia), or novelty attraction (neofilia). OK, you discuss this later (L 144).

L 92: I was wondering why you didn't use a binomial model, but there must be temporal autocorrelation in your timing events, so it is hard to know how many independent time events you would have, so the transformation seems indeed better.

L 94: activity is an interesting one. Why would activity influence habitat use by itself? More likely, more active fish are exposed to the different environments more, and therefore can choose better (which is probably why you demanded a minimum of four transitions). Or reverse, more active fish move between environments more, therefore not choosing. Either way, I think it is the interactions between activity*species, and activity*rearing environment, that are most interesting, not the main effect of activity.

L 94: you don't include side ("is the shallow light condition on the left or right") as a fixed factor?

L 94: what is trial number, and why does this have replicate values (required for a random effect)? Each trial supposedly is unique? If you refer to trial order, then this is not correct – fit it as a fixed effect.

L 96: you mention AIC, but then give p-values, which doesn't make much sense. And what are minimal adequate models – the outcome of stepwise elimination? Why not just give the results for the full models, testing all effects via loglikelihood ratio tests?

L 105: this results suggests you fitted trail number as a fixed effect, not as a random effect? And you tested an interaction, but you never specified before you would test any interactions – in “data analyses” indicate which interactions you included, and why.

L 105: 5 decimals for a p-value is way beyond its reliability: I think 3 decimals is precision enough.

L 116: I think this is a convoluted way to test an interaction?

L 132: genetic signatures?

L 138: and perhaps this depends also on competition? It is interesting that in the wild, the deep fish also use shallow waters, but not the reverse, mimicking the overall preference you detected here for shallow conditions perhaps? Might deep fish only use the deep when outcompeted by shallow fish in shallow waters (any field data to test this density- or frequency-dependence)? If so, and as you already mentioned, perhaps your design was not entirely appropriate to test your hypothesis, as reward was not included.

L 149: well, not quite – behavioural preference may depend on several factors (e.g. familiarity versus performance), and may shift over time.

L 151: no hypothesis of why all fish preferred shallow light conditions?

L 154: maybe add that future experiments should try to increase the importance of relative ecological performance across test environments, or something like that (i.e. greater reward/punishment).

Fig. 2: there seems to be a pattern here, I’m surprised it doesn’t come out? But it actually goes against your prediction, right? The deep species prefers shallow water more?

Supplementary materials:

L 21: there is a strong correlation between age and genetic background. Could the inclusion of age in your models have removed the effect of genetic background?

L 60: I’m not so convinced that average difference in light intensity isn’t important because cloud cover variation creates greater changes. Sun set creates even greater changes, but daylight spectrum differences still matter according to you. Perhaps acknowledge in your paper that this aspect was not manipulated, and might change the results. Someone else may try and replicate your study (in other species perhaps), taking this into account. Just as the reward issue.

I hope these comments help to further improve an already fine paper,
Pim Edelaar.

Author's Response to Decision Letter for (RSOS-181876.R0)

See Appendix A.

RSOS-181876.R1 (Revision)

Review form: Reviewer 1

Is the manuscript scientifically sound in its present form?

Yes

Are the interpretations and conclusions justified by the results?

Yes

Is the language acceptable?

Yes

Is it clear how to access all supporting data?

Yes

Do you have any ethical concerns with this paper?

No

Have you any concerns about statistical analyses in this paper?

No

Recommendation?

Accept as is

Comments to the Author(s)

The authors have done a thorough job in addressing my concerns and applying my suggestions. I have no further comments on the manuscript.

Review form: Reviewer 2 (Pim Edelaar)

Is the manuscript scientifically sound in its present form?

Yes

Are the interpretations and conclusions justified by the results?

Yes

Is the language acceptable?

Yes

Is it clear how to access all supporting data?

No

Do you have any ethical concerns with this paper?

No

Have you any concerns about statistical analyses in this paper?

No

Recommendation?

Accept with minor revision (please list in comments)

Comments to the Author(s)

I also was a reviewer for the previous version. The authors have responded well to the comments, and the paper has much improved. There are a few minor things that might still need some attention.

L 16: maybe say adaptive genetic differentiation

L 180: you still have repeat number as a random effect, so basically you are estimating a variance based on three values (and assuming the underlying distribution is normal, for no real good reason). This isn't really wrong, and it should not change the results much, but it makes more

sense to test it as a fixed effect. Especially since you are interested in the specific effect of each level: you later test whether repeat had a different effect for shallow-reared and deep-reared fish (L 192). So that is similar (but less elegant and less correct) to testing for a repeat*rearing environment effect, which you can easily include in your model if you include repeat as a fixed effect. I mentioned this before in my review, but the authors have not picked up on it (and did not understand the last comment about convoluted interaction testing - I hope it is clear now).

L 185: don't start a sentence with a number: Of the 80 trials, 75 were successful.

L 192: $p=0.407$? Yes, as in supplement.

L 215: perhaps also repeat that such a preference might be a contributing driver to adaptive population divergence in the first place, to make clear why this study is very interesting.

L 226: future experiments should provide *scope for* such reward ... Of course we should not reward the fish if they choose what we want ("predict") them to choose, but when the phenotype-environment match somehow affects their ecological performance (i.e. they reward themselves). For example, if food is provided on both sides, but they only see it on one side because of their visual capacities.

L 272: in visual traits and habitat in the wild do not

Decision letter (RSOS-181876.R1)

22-Feb-2019

Dear Mr Mameri:

On behalf of the Editors, I am pleased to inform you that your Manuscript RSOS-181876.R1 entitled "Visual adaptation and microhabitat choice in Lake Victoria cichlid fish" has been accepted for publication in Royal Society Open Science subject to minor revision in accordance with the referee suggestions. Please find the referees' comments at the end of this email.

The reviewers and Subject Editor have recommended publication, but also suggest some minor revisions to your manuscript. Therefore, I invite you to respond to the comments and revise your manuscript.

- Ethics statement

- Data accessibility

If you wish to submit your supporting data or code to Dryad (<http://datadryad.org/>), or modify your current submission to dryad, please use the following link:
<http://datadryad.org/submit?journalID=RSOS&manu=RSOS-181876.R1>

- **Competing interests**

- **Authors' contributions**

- **Acknowledgements**

- **Funding statement**

Because the schedule for publication is very tight, it is a condition of publication that you submit the revised version of your manuscript before 03-Mar-2019. Please note that the revision deadline will expire at 00.00am on this date. If you do not think you will be able to meet this date please let me know immediately.

When submitting your revised manuscript, you will be able to respond to the comments made by the referees and upload a file "Response to Referees" in "Section 6 - File Upload". You can use this to document any changes you make to the original manuscript. In order to expedite the

processing of the revised manuscript, please be as specific as possible in your response to the referees.

on behalf of Dr Kristina Sefc (Associate Editor) and Professor Kevin Padian (Subject Editor)
openscience@royalsociety.org

Reviewer comments to Author:

Reviewer: 1

Comments to the Author(s)

The authors have done a thorough job in addressing my concerns and applying my suggestions. I have no further comments on the manuscript.

Reviewer: 2

Comments to the Author(s)

I also was a reviewer for the previous version. The authors have responded well to the comments, and the paper has much improved. There are a few minor things that might still need some attention.

L 16: maybe say adaptive genetic differentiation

L 180: you still have repeat number as a random effect, so basically you are estimating a variance based on three values (and assuming the underlying distribution is normal, for no real good reason). This isn't really wrong, and it should not change the results much, but it makes more sense to test it as a fixed effect. Especially since you are interested in the specific effect of each level: you later test whether repeat had a different effect for shallow-reared and deep-reared fish (L 192). So that is similar (but less elegant and less correct) to testing for a repeat*rearing environment effect, which you can easily include in your model if you include repeat as a fixed effect. I mentioned this before in my review, but the authors have not picked up on it (and did not understand the last comment about convoluted interaction testing - I hope it is clear now).

L 185: don't start a sentence with a number: Of the 80 trials, 75 were successful.

L 192: $p=0.407$? Yes, as in supplement.

L 215: perhaps also repeat that such a preference might be a contributing driver to adaptive population divergence in the first place, to make clear why this study is very interesting.

L 226: future experiments should provide *scope for* such reward ... Of course we should not reward the fish if they choose what we want ("predict") them to choose, but when the phenotype-environment match somehow affects their ecological performance (i.e. they reward themselves). For example, if food is provided on both sides, but they only see it on one side because of their visual capacities.

L 272: in visual traits and habitat in the wild do not

Author's Response to Decision Letter for (RSOS-181876.R1)

See Appendix B.

Decision letter (RSOS-181876.R2)

05-Mar-2019

Dear Mr Mameri,

I am pleased to inform you that your manuscript entitled "Visual adaptation and microhabitat choice in Lake Victoria cichlid fish" is now accepted for publication in Royal Society Open Science.

on behalf of Dr Kristina Sefc (Associate Editor) and Kevin Padian (Subject Editor)
openscience@royalsociety.org

Associate Editor Comments to Author (Dr Kristina Sefc):
Associate Editor: 1
Comments to the Author:
(There are no comments.)

Reviewer comments to Author:

Appendix A

Reply letter to reviewers - RSOS manuscript submission (ref. RSOS-181876)

Dear Editorial Board and Reviewers,

We have now concluded the revision of our manuscript entitled “Visual adaptation and microhabitat choice in Lake Victoria cichlid fish”. We have considered all comments and suggestions by the referees, and we detail below how they were incorporated into the revision.

The main changes include adding information on opsin expression, moving supplementary methods to the main text (particularly details on light treatments), clarifying our data analyses and expanding our Discussion section to address specific explanations for our findings.

We believe the revised version of the manuscript has greatly improved in terms of contents and readability, and we hope that it now meets the standards for publication in *Royal Society Open Science*.

Daniel Mameri

REPLY TO REFEREE #1

Major concerns

Referee #1: In the wild, the species that occupies deeper water is exposed to a red-shifted spectrum but also to an overall lower light intensity. Without knowing the intensity of the two experimental light regimes, it is hard to conclude what aspect of the two light conditions drove the preference to the ‘broad spectrum’ condition. For example, it might be possible that fish actually preferred the brighter or dimmer light condition.

Reply: Indeed, the deep-water habitat is darker than the shallow-water habitat. This information was included in the supplementary information, but we have now incorporated it in the main ms (lines 146-150):

“At Python Islands, light intensity in the deeper P. sp. ‘nyererei-like’ habitat is about 34% of that in the shallow P. sp. ‘pundamilia-like’ habitat (Figure 1). We did not adjust the experimental light spectra to reproduce this difference (intensity in the deep condition was 70% of that in the shallow condition).”

It is very well possible that the behaviour of the fish, and their preference for either environment, was influenced by this difference light intensity. We did not address this possibility explicitly in the text, but we have now incorporated a few sentences in the Discussion (lines 260-267):

“Our light treatments differed in both spectral composition and intensity. Therefore, we cannot establish which of these aspects, or a combination of both, was responsible for the observed variation in visual habitat preference. Previous studies [20,24] have recorded fish preferences for darker or brighter environments, but these employed larger differences between habitats (ranging from two-fold to 20-fold) than we used in the present study (light intensity in the red-shifted condition was 70% of that in the broad-spectrum condition). Independently manipulating both spectral composition and light intensity is feasible, and would constitute a logical next step.”

Referee #1: What are the characteristics of the experimental light environment in terms of spectrum and intensity? Were light intensities matched to those encountered in the natural habitats? It would be useful to present these data relative to the lighting regimes encountered in the wild. What were the estimated quantum catches of the various opsin types expressed in each of the species?

Reply: Following the answer to the previous comment, we also moved to the main text the verification of the spectra by estimating photon capture (lines 140-145; the Figure is still presented in supplementary material, Figure S2).

Referee #1: One of the mechanisms of the presumed divergence in visual sensitivity between the species is differential opsin gene expression. However, the authors do not provide any evidence on whether opsin expression differed between the tested species. Is it possible that all fish exhibited similar gene expression as a result of their exposure to the same light condition prior to the beginning of the experiment? How fast such plasticity in opsin gene expression could be?

The authors state (lines 55-57) that they manipulated visual development by raising the fish under different light conditions. However, they provide no evidence for the capacity of such manipulation. In other words, did the different light conditions actually affected the visual sensitivity, and if so, how?

Reply: Opsin expression was not measured in the individuals used for this study. However, we have documented changes in opsin expression in other individuals raised in these same light treatments. We now incorporate this information in the text:

Introduction: (lines 61-63): *“We have previously shown that these light treatments induce changes in opsin expression in Pundamilia [15]. We therefore predict that the light regime during development influences visual habitat preference as well.”*

Discussion (lines 246-252): *“We have previously shown that the light treatments we used here induce changes in opsin expression in Pundamilia [15]: deep-reared fish express less short-wavelength-sensitive opsin (SWS2a) and more long-wavelength-sensitive opsin (LWS). Yet, we did not observe a sustained effect of the rearing environment on preference. Possibly, the induced changes in opsin expression (5-15%) were too subtle to cause behavioural effects. More extreme rearing environments can generate larger changes (e.g. [25]) that would potentially influence visual habitat preference in a more persistent way.”*

We do not know how fast opsin expression can change in response to altered light conditions. We speculate that it takes longer than the 1-hour duration of the experiments, but we cannot rule out that exposure to the experimental conditions had an effect.

We have added a section to the Discussion to address these points (lines 253-259):

“An alternative explanation for the lack of a sustained effect of the rearing light treatment, is that the differences in opsin expression were erased during the experimental trials, as fish were exposed to both light conditions during this time. It is unknown how quickly opsin expression can change in response to altered light conditions. Studies in killifish [26] and cichlids [27] suggest that changes can occur within a few days, but most studies have used much longer exposure times (weeks to months). We expect that exposure of 1 hour is too short to induce significant changes, but we cannot rule this out.”

Minor concerns

Referee #1: Lines 39-43 – It is not clear what are the differences between *P. pundamilia* and *P. sp. 'pundamilia-like'*. The same is also true for *P. nyererei*. Please clarify.

Reply: We have expanded the text on the subject species, explaining the nomenclature and summarising the current knowledge on their evolutionary history (lines 73-88). Specifically in lines 83-88, we wrote:

“Here, we used laboratory-bred offspring from wild-caught fish, collected in 2010 and 2014 at Python Islands in southern Lake Victoria [6]. Until recently, all red populations were thought to belong to P. nyererei and all blue populations to P. pundamilia. However, the populations in the western and southern Mwanza Gulf (including Python Islands) represent a separate speciation event and are referred to as P. sp. 'pundamilia-like' & P. sp. 'nyererei-like' [10].”

Referee #1: Line 46 – ‘They’ presumably refers to the two species, but this isn’t clear enough.

Reply: Changed accordingly: “They” replaced with “The two species”.

Referee #1: Line 51 – Please elaborate on how the causality between visual sensitivity and habitat preferences is being determined.

Reply: This is explained later in lines 59-61:

“(…) to assess the causal relationship between visual sensitivity and habitat preference, we manipulate visual development by raising the fish under different light conditions.”

To further clarify this, we have now added the information that this manipulation is known to induce changes in opsin expression (see above) and we have clarified the method of splitting families between treatments (lines 92-97):

“To control visual development of the test fish, they were removed as eggs from mouth-brooding females around the time of hatching (~5 days after fertilization). Each family was then split, divided equally over two light treatments that mimicked the natural light environments experienced by P. sp. 'pundamilia-like' (shallow water, 0-2 m) and P. sp. 'nyererei-like' (deeper water, 0-5 m) at Python Islands [6] (Figure 1; details below).”

Referee #1: Line 74 – ‘only group-level data was recorded’ should be ‘only group-level data were recorded’.

Reply: Changed accordingly: “was” replaced by “were”.

Referee #1: Line 74-75 – What was the light regime before fish were exposed to the experimental light regimes?

Reply: Fish were reared in the same light conditions as those used in the experiment. Thus, half of the groups were reared in the broad-spectrum condition, and the other half was reared in the red-shifted light condition. They remained in these conditions throughout the experimental period, and only experienced the other condition during the experiment itself. Thus, if reared in broad-spectrum light, fish were never exposed to the red-shifted light condition (and vice-versa). We have added a few sentences to make this clear (lines 97-100):

“They remained in these conditions throughout the experimental period, and only experienced the other condition during the experiment itself. Thus, fish reared in broad-spectrum light were never exposed to the red-shifted light condition (and vice-versa), until testing.”

Referee #1: Line 86-88 – “After analysing the first two trials, we submitted *P. sp.* ‘pundamilia-like’ and *P. sp.* ‘nyererei-like’ to a third trial to increase statistical power for testing species differences.” Why not simply say that groups were tested three times. Did the third trial differ from the two previous ones? Actually, only when I reached Figure 2 I realized that the third trial differed from the rest in that hybrids were not tested. It might be better to clarify this issue up front.

Reply: Agreed and clarified in lines 168-172:

*“All groups were tested twice, with approximately 2 weeks between repeats. Light environments were switched between tank sides after the first repeat. After analysing the first two repeats, we submitted *P. sp.* ‘pundamilia-like’ and *P. sp.* ‘nyererei-like’ groups (but not hybrid groups) to a third repeat to increase statistical power for testing species differences.”*

Referee #1: Line 99 – There is inconsistency in the use of the term ‘trial’. Line 87 says that the species were subjected to a third trial. However, in line 99, it is indicated that ‘75 of the 80 trials were successful’. Please clarify.

Reply: The meaning of ‘trial’ is the same, but in line 87 we are referring to a third trial from the perspective of the test group (i.e. the 4 fish). We have clarified this by using the term ‘repeat’ instead of trial, and by inserting ‘group’ in lines 168-172.

Referee #1: Line 119 – ‘nor trials’ should probably be replaced with ‘or trials’.

Reply: Changed accordingly: “nor” replaced by “or”. We also replaced ‘trial’ with ‘repeat’ to clarify that this is the trial number as experienced by the tested group.

REPLY TO REFEREE #2

Referee #2: The experimental design is adequate, as are the analysis. Sample size is perhaps somewhat small (as in many behavioural studies), which may mask small effects. It might be worth to discuss the issue of lack of power, and whether any observed trends at least go in the predicted direction or not. I also have a few suggestions for improvement, or clarification, of analyses. The main predictions are mostly not confirmed. While I’m aware that conceptually negative results are just as good as positive ones and would not want to promote a publication bias in favour of positive results, there is always the possibility that the hypotheses are in fact correct, but the experimental test was not. You discuss that perhaps reward for correct habitat choice was not sufficiently in place, which is important. You discard the difference in light intensity across habitats which was not implemented, for reasons that I didn’t find so convincing. And then there may be another 1000 factors that are important for the fish, but that we are not aware of and therefore were not (correctly) manipulated. Perhaps in your paper, mention this possibility – that your experimental test might just not have created the ecological setting that does play out in the field, and therefore that the results are limited. This is fine, that is how science works, and perhaps the hypotheses are indeed wrong. But if you had achieved confirmatory results, the ecological relevance would probably not have to be questioned to the same extent.

Reply: We agree that the lack of a reward, and the difference in light intensity between treatments, may have influenced the results, and we did not discuss these issues extensively in the manuscript. We have adjusted this by moving some of the supplementary information to the main manuscript and expanding the text in the Discussion. Please see replies below for more details on each of these issues.

Referee #2: 32: add: and thereby increase their ecological performance

Reply: Changed accordingly.

Referee #2: L 60: is there nothing known about (co)dominance in expression?

Reply: There is no published information about this. However, our own data suggest that heterozygotes for the LWS allele (i.e. the genotype of most hybrids) express both alleles. At this stage, we do not think that this dataset is large enough to warrant mentioning it in this manuscript, and we prefer to keep this conservative statement about hybrids having ‘presumably intermediate characteristics’.

Referee #2: L 61: as written now, your predictions appear exclusive, which cannot be: they prefer their natural light regime (L 58), and they prefer their rearing light regime. Why not simply state that you will test if, and to what extent, genetic background and rearing regime affect habitat preference? (And then mention the directions re. these effects).

Reply: We agree that this might have been confusing; we have rewritten this section (lines 56-63):

*“If genetic differences determine visual habitat preference, we predict that individuals of either species prefer the light regime that is closest to the one their populations are adapted to. In addition to genetic differences however, developmental plasticity may contribute to variation in visual sensitivity [14]. To explore this, and to assess the causal relationship between visual sensitivity and habitat preference, we manipulate visual development by raising the fish under different light conditions. We have previously shown that these light treatments induce changes in opsin expression in *Pundamilia* [15]. We therefore predict that the light regime during development influences visual habitat preference as well.”*

Referee #2: L 68: so did you allocate entire families to treatment, or did you split families between treatments (a stronger design)?

Reply: We did the latter; we split families between treatments. We have adjusted the text to make this more clear (i.e. moved information from supplementary text to main text; lines 92-100):

*“To control visual development of the test fish, they were removed as eggs from mouth-brooding females around the time of hatching (~5 days after fertilization). Each family was then split, divided equally over two light treatments that mimicked the natural light environments experienced by *P. sp. 'pundamilia-like'* (shallow water, 0-2 m) and *P. sp. 'nyererei-like'* (deeper water, 0-5 m) at Python Islands [6] (Figure 1; details below). They remained in these conditions throughout the experimental period, and only experienced the other condition during the experiment itself. Thus, fish reared in broad-spectrum light were never exposed to the red-shifted light condition (and vice-versa), until testing.”*

Referee #2: L 71: is the range 0-5 m the natural range? How do you expose fish to the light of a depth range? It is in the supplement (weighting by depth use), but briefly clarify this here.

Reply: We have moved this information to the main text (Figure 1 and lines 130-139):

*“For each measurement series separately we then estimated the light environments experienced by *P. sp. 'pundamilia-like'* and *P. sp. 'nyererei-like'*, by calculating a weighted average of the spectra at each depth, using as a weighting factor the depth distribution of each species as*

reported in [6]. The average of the 4 resulting species-specific light spectra was mimicked in the laboratory by halogen lights (Philips Halogen Masterline ES, 30 and 35W) filtered with a green filter (#243 by LEE, Andover, UK). In the shallow light condition, blue lights (Paulmann 88090 ESL Blue Spiral 15W) were added to create a broad light spectrum. In the deep light condition, short wavelength light was reduced by adding a yellow filter (LEE #15). The resulting downwelling irradiance was measured with the same equipment as in the field (two tanks for each light condition, 4 positions in each tank)."

Referee #2: L 75: this is important, since you might then be testing for novelty avoidance (neofobia), or novelty attraction (neofilia). OK, you discuss this later (L 144).

Referee #2: L 92: I was wondering why you didn't use a binomial model, but there must be temporal autocorrelation in your timing events, so it is hard to know how many independent time event you would have, so the transformation seems indeed better.

Referee #2: L 94: activity is an interesting one. Why would activity influence habitat use by itself? More likely, more active fish are exposed to the different environments more, and therefore can choose better (which is probably why you demanded a minimum of four transitions). Or reverse, more active fish move between environments more, therefore not choosing. Either way, I think it is the interactions between activity*species, and activity*rearing environment, that are most interesting, not the main effect of activity.

Reply: The results indicate that more active fish have weaker preferences. Of course it is difficult to establish the direction of causality here, but we think that a lack of a strong preference promotes more frequent switching between environments. Importantly, fish activity did not differ between species groups, rearing conditions and repeats. Also, the overall preference for the broad-spectrum light condition, and the initial effect of the rearing condition, were present irrespective of fish activity.

Referee #2: L 94: you don't include side ("is the shallow light condition on the left or right") as a fixed factor? L 94: what is trial number, and why does this have replicate values (required for a random effect)? Each trial supposedly is unique? If you refer to trial order, then this is not correct – fit it as a fixed effect.

Reply: The variable "trial number" (term replaced by "repeat") refers to the round of trials for each fish group (varies between 1-first round, 2-second round and 3-third round, hybrids excluded in the last one). Between trial round 1 and 2, we switched the light environment on each side of the tank (shallow from left to right side, and deep from right to left), to account for any side effects (though no side effects were found in preliminary studies using the same tank).

Referee #2: L 96: you mention AIC, but then give p-values, which doesn't make much sense. And what are minimal adequate models – the outcome of stepwise elimination? Why not just give the results for the full models, testing all effects via loglikelihood ratio tests?

Reply: We agree that testing all effects via loglikelihood ratio tests and other methods such as Satterthwaite's ANOVA are more straightforward. We re-did the analyses using linear mixed models combined with Satterthwaite's ANOVA, and re-wrote the data analyses section to better clarify our methodology. Additionally, we added our models to the Supplementary Results (Tables S2-S4). Please see our reply to the next comment for further details.

L 105: this results suggests you fitted trail number as a fixed effect, not as a random effect? And you tested an interaction, but you never specified before you would test any interactions – in “data analyses” indicate which interactions you included, and why.

Reply: Indeed our methodology for the data analyses was not clear. The linear mixed model approach we followed, with the inclusion of ‘repeat’ as random factor, was not appropriate to test the effect of this variable with fixed effects. Furthermore, the lack of statistical power to test interactions should be taken into account to avoid misleading results. We clarified our data analyses section in that sense, and applied linear mixed models (presented in Tables S2-S4) for each of the variables we wanted to evaluate: preference for the shallow light condition (Table S2), preference for the rearing environment (Table S3) and preference strength (Table S4). As we were interested in seeing if preference for the rearing environment would vary among repeats in both shallow and deep-reared fish, we included ‘repeat’ as a fixed factor in the model presented in Table S3. The section concerning data analyses was re-written as follows (lines 174-182):

*“We calculated the proportion of time spent on the side of the tank with shallow light conditions (summed for the individuals in a group) as a measure of preference, ranging from 0 to 1, and then applied an arc-sin transformation to improve data distribution. This preference score was fit into a linear mixed model, with the genetic group (*P. sp.* ‘pundamilia-like’, *P. sp.* ‘nyererei-like’ and hybrids), rearing environment (shallow and deep), age and activity as fixed effects. Random effects included repeat number, nested in fish group, nested in family (some groups came from the same family – see Table S1). Significance of each variable was tested using Satterthwaite’s ANOVA, and the minimum adequate model was obtained by removing non-significant variables following a stepwise approach.”*

Referee #2: L 105: 5 decimals for a p-value is way beyond its reliability: I think 3 decimals is precision enough.

Reply: Agreed and changed accordingly (all p-values with 3 decimals).

L 116: I think this is a convoluted way to test an interaction?

Reply: We do not understand this comment. Perhaps the reviewer refers to lines 193-194, where we test the species effect again by excluding the hybrids. This is just to show that there really is no difference.

Referee #2: L 132: genetic signatures?

Reply: Changed accordingly: “genetic signatures of divergent selection”.

Referee #2: L 138: and perhaps this depends also on competition? It is interesting that in the wild, the deep fish also use shallow waters, but not the reverse, mimicking the overall preference you detected here for shallow conditions perhaps? Might deep fish only use the deep when outcompeted by shallow fish in shallow waters (any field data to test this density- or frequency-dependence?)? If so, and as you already mentioned, perhaps your design was not entirely appropriate to test your hypothesis, as reward was not included.

Referee #2: L 151: no hypothesis of why all fish preferred shallow light conditions?

Reply: We have expanded the Discussion to address these comments; we now explicitly discuss the overall preference for the shallow environment and the role of competition and predation (see below).

Referee #2: Fig. 2: there seems to be a pattern here, I'm surprised it doesn't come out? But it actually goes against your prediction, right? The deep species prefers shallow water more?

Reply: Indeed, this goes against the prediction that both species would prefer their natural habitat. We have now added a few sentences to address the overall preference for the broad-spectrum light condition, including the role of competition and predation (lines 226-233):

“For P. sp. 'pundamilia-like', the observed preference for the broad-spectrum light condition was expected, as it corresponds to its natural habitat. However, we expected P. sp. 'nyererei-like' to prefer the red-shifted light condition. P. sp. 'nyererei-like' also occurs in shallow water, but it is most abundant in deeper waters [6,11]. Possibly, this depth distribution does not reflect the preferred environment for this species, but rather emerges from ecological interactions such as competition and predation. Predation risk by birds is higher in shallow waters, which might affect especially the brightly coloured males of P. sp. 'nyererei-like' [23]. Both of these factors, competition and predation, were absent in our experiment.”

Referee #2: L 149: well, not quite – behavioural preference may depend on several factors (e.g. familiarity versus performance), and may shift over time.

Reply: Adjusted accordingly (lines 268-269):

“To conclude, in this study we find evidence that Pundamilia cichlid fish exert significant preference for visual habitat, preferring broad-spectrum over darker, red-shifted light conditions.”

Referee #2: L 154: maybe add that future experiments should try to increase the importance of relative ecological performance across test environments, or something like that (i.e. greater reward/punishment).

Reply: In lines 234-236, we mention:

“Possibly, our fish did not have enough opportunity to evaluate their performance in the two environments: while we added food cues to stimulate exploration, we did not provide an actual reward. Matching habitat choice may be most pronounced when it generates substantial reward [22]. Therefore, future experiments should provide such reward, in the form of e.g. food or social interaction.”

Supplementary materials

Referee #2: L 21: there is a strong correlation between age and genetic background. Could the inclusion of age in your models have removed the effect of genetic background?

Reply: Indeed, the different species groups differ in age, although they are all subadults. However, the factor 'age' did not significantly affect preference (Table S2).

Referee #2: L 60: I'm not so convinced that average difference in light intensity isn't important because cloud cover variation creates greater changes. Sun set creates even greater changes, but daylight spectrum differences still matter according to you. Perhaps acknowledge in your paper that this aspect was not manipulated, and might change the results. Someone else may try and replicate your study (in other species perhaps), taking this into account. Just as the reward issue.

Reply: Also in response to Referee 1, we have adjusted the text to address this more thoroughly, and to make it clear how light intensities differed, both in the experiment and in nature. We

have moved the methodological information from supplementary materials to the main text. Moreover, we agree that differences in light intensity may have influenced the findings. We did not address this possibility explicitly in the original text, but we have now incorporated a few sentences in the Discussion, such as in lines 260-267:

“Our light treatments differed in both spectral composition and intensity. Therefore, we cannot establish which of these aspects, or a combination of both, was responsible for the observed variation in visual habitat preference. Previous studies [20,24] have recorded fish preferences for darker or brighter environments, but these employed larger differences between habitats (ranging from two-fold to 20-fold) than we used in the present study (light intensity in the red-shifted condition was 70% of that in the broad-spectrum condition). Independently manipulating both spectral composition and light intensity is feasible, and would constitute a logical next step.”

I hope these comments help to further improve an already fine paper,

Reply: Thank you!

Appendix B

Reply letter #2 to reviewers - RSOS manuscript submission (ref. RSOS-181876)

Dear Editorial Board and Reviewers,

We have concluded the revision of our manuscript entitled “Visual adaptation and microhabitat choice in Lake Victoria cichlid fish”, accepted for publication under minor revisions. We have considered the remaining comments and suggestions by the referees, and we detail below how they were incorporated into the manuscript.

We would like to thank all the suggestions made by the two referees during the revision process, and also the editors involved in our manuscript submission to *Royal Society Open Science*.

Daniel Mameri

Reviewer: 1

Comments to the Author(s)

The authors have done a thorough job in addressing my concerns and applying my suggestions. I have no further comments on the manuscript.

Reply: We thank you for all your suggestions that greatly improved the manuscript.

Reviewer: 2

Referee #2: I also was a reviewer for the previous version. The authors have responded well to the comments, and the paper has much improved. There are a few minor things that might still need some attention.

Reply: We thank you for all your suggestions that greatly improved the manuscript. We tried to address the following issues below.

Referee #2: L 16: maybe say adaptive genetic differentiation

Reply: We appreciate this suggestion, but decided to not change this. The reason is that part of the genetic differentiation resulting from divergent habitat choice will not be adaptive (neutral, perhaps even maladaptive). The reduced interbreeding will affect loci all over the genome.

Referee #2: L 180: you still have repeat number as a random effect, so basically you are estimating a variance based on three values (and assuming the underlying distribution is normal, for no real good reason). This isn't really wrong, and it

should not change the results much, but it makes more sense to test it as a fixed effect. Especially since you are interested in the specific effect of each level: you later test whether repeat had a different effect for shallow-reared and deep-reared fish (L 192). So that is similar (but less elegant and less correct) to testing for a repeat*rearing environment effect, which you can easily include in your model if you include repeat as a fixed effect. I mentioned this before in my review, but the authors have not picked up on it (and did not understand the last comment about convoluted interaction testing ? I hope it is clear now).

Reply: We thank you for the clarification. We had three repeats in our experimental design, and while *P. pundamilia* and *P. nyererei* groups were tested three times, hybrids were only tested twice. Thus, we cannot adequately assess the effect of ‘repeat’ in the full model that includes all fish. We do need to control for it though. We therefore believe that nesting repeat as a random effect in fish group and fish family (all random effects) for the overall model (Table S2), in which we test our main hypothesis, seems the best approach. We agree that the way we presented the effect of repeat for shallow and deep-reared groups (now using ‘repeat’ as a fixed factor) in Table S3 may not be the most straightforward approach, but considering the data we had, it should be suitable. Also, we believe our sample sizes are not sufficient to explore interaction terms in a mixed model in a way that F and p-values are fully reliable, so we preferred to present only the main effect of ‘repeat’ in Table S3.

Referee #2: L 185: don't start a sentence with a number: Of the 80 trials, 75 were successful.

Reply: Changed accordingly: “Of the 80 trials, 75 were successful.” (line 185).
L 192: $p=0.407$? Yes, as in supplement.

Reply: Changed accordingly: “ $p=0.407$ ”, a ‘0.’ was indeed missing (line 192).

Referee #2: L 215: perhaps also repeat that such a preference might be a contributing driver to adaptive population divergence in the first place, to make clear why this study is very interesting.

Reply: This is a good suggestion. We added “Such preferences could contribute to genetic differentiation between populations, particularly during the early stages of adaptive divergence.”

Referee #2: L 226: future experiments should provide *scope for* such reward ... Of course we should not reward the fish if they choose what we want (?predict?) them to choose, but when the phenotype-environment match somehow affects their ecological performance (i.e. they reward themselves). For example, if food is provided on both sides, but they only see it on one side because of their visual capacities.

Reply: We agree and clarified as follows (lines 226-231):

“Possibly, our fish did not have enough opportunity to evaluate their performance in the two environments: while we added food cues to stimulate exploration, we did not provide an actual reward that could generate vision-dependent variation in performance. Matching habitat choice may be most pronounced when it generates a substantial advantage [22]. Therefore, future experiments should provide opportunity for such an advantage, in the form of e.g. food reward or social interaction.”

Referee #2: L 272: in visual traits and habitat in the wild do not

Reply: Adjusted accordingly (lines 275-276): *“Species differences in visual traits and habitat in the wild do not translate into differences in preference.”*